# Effect of Attainment Value and Positive Thinking as Moderators of Employee Engagement and Innovative Work Behaviour

**Peerapong Pukkeeree [1], Khahan Na-Nan [2],* and Natthaya Wongsuwan [3]**

1    Faculty of Political Sciences, Ramkhamhaeng University, Bangkok 10240, Thailand; puk_peerapong@ru.ac.th
2    Faculty of Business Administration, Rajamangala University of Technology Thanyaburi,
     Thanyaburi 12110, Thailand
3    Faculty of Business Administration, Ramkhamhaeng University, Bangkok 10240, Thailand;
     natthaya@ru.ac.th
*    Correspondence: khahan_n@rmutt.ac.th

**Abstract:** Influences of attainment value and positive thinking were assessed as moderators of employee engagement and innovative work behaviour. A cross-sectional design was utilised with questionnaires submitted to 348 human resource officers to test the proposed relationships. SPSS 21 and PROCESS macro 3.1 were used for statistical analysis. Results revealed that positive thinking effectively moderated attainment value and employee engagement with regard to innovative work behaviour with statistical significance. Results can be utilised by managers and human resource departments to promote and support innovative work behaviour. Moreover, employees should be encouraged and motivated to perceive attainment value through positive thinking. Findings contribute to the literature on employee engagement and innovative work behaviour by highlighting that attainment value and positive thinking act as moderators that promote employee engagement and innovative work behaviour.

**Keywords:** employee engagement; attainment value; positive thinking; innovative work behaviour

## 1. Introduction

Innovative work behaviour (IWB) is important for the progress of people, workplaces, organisations and economy in countries which have to adapt to changes and competition, as well as to increase industrial product values [1]. Companies that successfully build innovation are usually supported, promoted and motivated by the creative innovation of their staff [2]. Rowley, et al. [3] and Tidd and Bessant [4] suggested that promoting and supporting creative innovation in the workforce should be highly valued as a human resource investment. Similarly, Sattabut [5] stated that creative thinking is the ability of people to originate innovation when performing tasks. Companies that promote successful innovative creation view their employees as important resources of original and inventive performance. Therefore, examining the innovative work performance of employees is necessary for companies that are currently facing increasingly rapid changes [6]. Employee engagement (EE) is another motivation to maximise the abilities of staff to perform proactive tasks as IWB [7]. High levels of EE will increase innovative performance levels of employees, leading to company growth [8,9]. Rothmann and Rothmann Jr. [10] and Slåtten and Mehmetoglu [11] considered that EE theory concisely described employees' characteristics to work collaboratively with full potential to conduct their responsibilities effectively and respond to customers' satisfaction both within and outside the organisation. However, IWB may depend on other factors. Theoretical observations made in previous studies determined large coefficient effects (more than 0.20) between EE and IWB.

Chin [12] stated that large coefficient effects of more than 0.20 were probably influenced by other variables. Perceived attainment value (AV) is another factor that encourages employees to create high-quality work since they perceive future success if they fulfil their responsibilities effectively [13,14]. Moreover, according to the broaden-and-build theory, positive thinking (PT) can stimulate feelings through creative thought that then develops into behavioural repertoire to build specialised skills [15,16]. For both academics and researchers, the question arises whether factors of AV and PT are possible moderators of EE and IWB. Interviews with experts in human resources and behavioural science suggest that perceived AV and PT are aspects of people's emotions or feelings as moderators which stimulate or motivate EE, resulting in increasing employees' IWB [17].

Research results concurred with Kapikiran [18] and concluded that harmony emanates from positive thinking (PT) as an essential facet to regulate both personal and job satisfaction. Moreover, Chang, et al. [19] stated that focusing on PT promotes personal perspective as a starting point to respond to problems or issues. Adopting PT increases personal readiness for task implementation to meet the job requirements and attain the expected outcome. The perceived difficulty of a task is conceptualised as a framework related to the cause and effect variables of the challenge and value of the job. Consideration of the task properties will determine the appropriate setting of goals and the enthusiasm to perform efficiently and successfully realise the objectives. Arieli, et al. [20] and Purc and Laguna [14] highlighted key issues concerning attainment perception in relation to various personal innovative work behaviours, while Schwartz [21] considered that attainment perception could be used to accurately predict an employee's attainment value (AV).

To examine these concepts and theories, previous studies and recommendations from educators and researchers were collated with two main research objectives: (i) to study the effects of EE on IWB and (ii) to study AV and PT as moderators of EE and IWB.

## 2. Literature Review and Research Framework

### 2.1. Relationship between EE and IWB

Kahn [22] proposed the engagement theory, first published in the *Academy of Management Journal*, in his paper entitled Psychological Conditions of Personal Engagement and Disengagement at Work. His results suggested that EE occurs from people's psychological state in relation to their job responsibilities in three aspects of meaningfulness, safety and availability. All of these aspects are components or main indicators of EE. In other words, employees demonstrate IWB when their psychological state recognises that their job is meaningful to both themselves and their employer. They perform tasks with safety and security when the necessary job components are available. Later, the engagement theory was further studied and developed by Gubman [23], Hewitt Associates [24], May, et al. [25], Richman [26], Saks [27], Shaw [28], Soane, et al. [29] and Potoski and Callery [30]. They defined EE as behavioural and emotional expressions of work conditions rather than set roles or responsibilities. EE also includes the expression of work ownership and the desire to achieve targets. Schaufeli, et al. [31] expressed EE in three ways; as (1) vigour referring to work behaviour expression with psychological resilience and persistence when facing various problems and obstacles, (2) dedication referring to attitudes of willingness, pride, inspiration and challenges to work assignments and (3) absorption referring to engagement, loyalty and happiness to perform responsibility.

The concept of IWB refers to employees who aim to initiate new things and introduce new ideas useful for the production of products, services and new work procedures to achieve targets [32–36]. Similarly, Yeoh and Mahmood [37] stated that employees with IWB provide creative ideas concerning work procedures, methods, products and services. Improved work behaviour that attains targets leads to new initiatives and the introduction of novel ideas [38]. The expressive dimension of IWB closely relates to creative behaviour, especially by thinking up new useful ideas [39]. Nevertheless, IWB differs

from creative thinking since it is aimed at applying new ideas that are useful for oneself, departments or organisations.

Meanwhile, Caniëls and Veld [40] suggested that innovative work behaviour (IWB) had a significant statistical relationship with high work performance, whereas Bysted [41], Bos-Nehles, et al. [42], Jol, et al. [43], Shanker, et al. [44] and Veenendaal and Bondarouk [45] revealed the significance of comparing the contexts of formal development. They found that individual intuition led to the creation and development of a strategic perspective that can be transferred to the work concept in groups and organisations.

Employee engagement has a positive influence on performance [46] and innovation both in qualitative and quantitative dimensions. Sundaray [47] indicated that EE positively relates to an employee's work performance, creative thinking and innovation, while Amabile [39] proposed creative thinking theory and explained that EE and innovation closely relate to each other. Similarly, Csikszentmihalyi and Csikszentmihalyi [48] stated that creative thinking is formed through interaction of the engagement, perception or PT of employees. EE with high degrees of vigour, dedication and absorption enhances and supports employees' behaviours to create innovative work with full dedication and willingness [49]. Vithayaporn and Ashton [50] found that EE could predict up to 75.00% variance of IWB. According to the concepts, theories and research findings mentioned above, the first hypothesis was posited as follows:

**Hypothesis 1 (H1).** *Employee engagement influences innovative work behaviour.*

*2.2. Relationship between AV and EE on IWB*

Expectancy-value theories believe that "One action produces particular results". If a person believes that some results are likely to occur, then he/she will adopt behaviour to obtain the expected results. Expectancy-value theories consist of two components: expectancy for success and subjective task values. Expectancy determines a person's action leading to the required or desirable practices. Many factors affect whether a person wants to complete an action, such as working hard to meet an important deadline. Conversely, subjective value is a person's belief in the importance of obtaining results. Performance–outcome expectancy explains the perception of the relationship between an action and its results. The nature of interest in the given activity is known as the intrinsic value and the importance of success, known as AV, is the firm belief in achieving the desired result. Performance–outcome expectancy is also called instrumentality as the perceived utility value [51–55].

Schwartz [21] mentioned that perceived AV such as power, achievement, motivation, comfort, safety, friendship and job direction is a moderator relating to work performance. If perceived AV is high, this will relate to an employee's work performance. Arieli, Sagiv and Roccas [20] stated that if employees perceive self-value, they will devote their effort and willingness to work effectively. Similarly, Purc and Laguna [14] noted that an increased level of an employee's AV perception will result in a high level of IWB. Theories and empirical research suggest that perceived AV is a moderator between EE and IWB. Cartwright and Holmes [56], Macey and Schneider [57], Soane, Truss, Alfes, Shantz, Rees and Gatenby [29] and Seligman [58] contributed to the notion that PT has a significant relationship with employee engagement (EE) and prediction of successful task accomplishment. As a result, the second hypothesis was posited as follows:

**Hypothesis 2 (H2).** *Perceived attainment value moderates employee engagement and innovative work behaviour.*

*2.3. Relationship between PT as a Moderator of AV and EE on IWB*

This research examines and extends this assumption for a better explanation, understanding and restructuring following the broaden-and-build theory. This theory suggests that positive emotions (enjoyment, happiness, joy, interest and anticipation) broaden one's awareness and encourage novel,

varied and exploratory thoughts and actions. This broadened behavioural repertoire builds skills, while curiosity about a landscape becomes valuable navigational knowledge, pleasant interactions with a stranger become a supportive friendship and aimless physical play becomes exercise and physical excellence, expectation, interest, intuition, interpretation, integration and institutionalisation [16]. PT is a thought process formed by perception and interpretation to generate attitudes and consciousness for oneself, others, objects or situations. PT is a method for tolerating and encountering problems with the strength to live happily and successfully [59,60]. PT is similar to optimism, whereby a person accepts negative reality with understanding, sees problems or difficulties as things that usually happen and knows how to gain advantages from the positive side of such situations. Bandansin [61] stated that PT is a form of human thinking which is creative and useful for themselves and societies. PT includes thinking up new inventions and new solutions. These ideas are advanced skills and the working process from the brain includes initiatives such as quick, fluent, flexible and creative thinking. Creative thinkers conjure up new ideas from existing data by relating the concepts differently. Therefore, PT can be both a thinking process and a consequence of a person's thinking. It is formed by positive perception and cognition; belief in possibilities based on reality, logics and generosity; being determined, self-confident and able to control emotions. PT results in good and appropriate behaviour [62].

Khan and Husain [63] found that PT performs as a moderator in the relationship between independent variables and dependent variables with statistical significance, while Bandansin [61] indicated that PT was helpful for nurses to express their IWB and provide effective services for patients. These nurses integrated their existing knowledge with PT to form their IWB. As a result, the patients were increasingly impressed with their services or IWB. Supannopaph [62] mentioned that PT directs a person to perceive, behave and create positive results such as a healthy mind, opportunities for success, self-confidence, optimism and being realistic. Hazelton [64], Kang and Sung [65], Tufail, et al. [66] and Young, et al. [67] conducted quantitative research and concluded that PT had a statistically significant relationship with EE. Their empirical findings supported that PT improved EE. According to these concepts, theories and previous research, the third hypothesis was posited as follows:

**Hypothesis 3 (H3).** *Positive thinking and attainment value combine as moderators in the relationship between employee engagement and innovative work behaviour.*

### 2.4. Research Framework and Research Method

The empirical research on EE proved that EE became a substantial factor that affects IWB. AV and PT are the moderators in correlation to EE and IWB. Meanwhile, AV and PT were analysed and synthesised in the study framework shown in Figure 1.

Analysing units of this research included human resource (HR) officers who were members of the Personnel Management Association of Thailand. Selection of HR officers as the targeted population was appropriate because they were expected to express IWB to solve various problems and manage and develop effective employees. HR officers usually possess EE because they work with people who have various problems; without EE they cannot perform their work effectively. HR officers also know their career paths and targets. Therefore, they are people with perceived AV which companies push their employees to possess. PT is an important factor when working with other people as well as for performing responsibilities dealing with complicated and challenging problems. Therefore, HR officers must have positive attitudes towards their responsibilities and interactions with other employees.

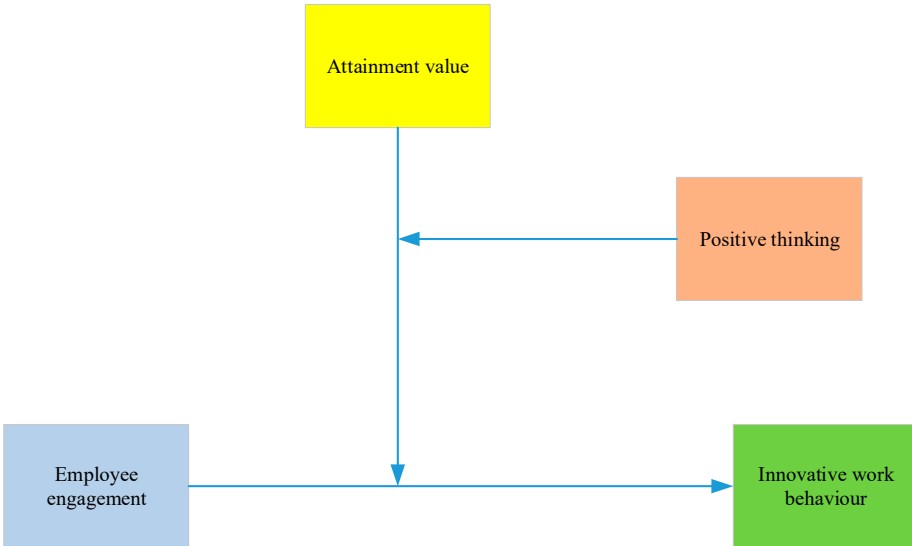

**Figure 1.** Conceptual research framework.

The sample size included 400 sampling units. A convenience sampling method was chosen using email correspondence as the most effective way to support the variety and quality of the respondents. Online surveys have both advantages and disadvantages, but responses can be obtained quickly without geographical limitation. The email surveys were sent to 400 selected samples. Within 4 weeks, 253 surveys were returned, so another 147 email surveys were sent out to a new group of samples and 95 responding emails were obtained within a similar waiting period of 4 weeks. In total, there were 348 respondents at 87% of the sample size. Joungtrakul [68] stated that a good response rate should be more than 70%, whereas Fincham [69] suggested that a response rate over 80% could be regarded as acceptable at a high level. Therefore, the response rate was good and sufficient for further analysis.

## 3. Validity, Reliability and Respond-Bias Control

### 3.1. Validity and Reliability of the Instrument

This research design followed a comparative longitudinal case study approach as the most appropriate method to identify the main sequences of sampling population and culture for different units of analysis in the Thai context. Data collection for EE was adapted from Saks [27] and comprised 6 items including "you feel that the job is part of your life", "although the work takes a lot of time to complete you like to do it" and "you tend to focus on the work without thinking of anything else". Data collection for IWB was adapted from Janssen [32] and included 4 items; for example, "you often develop new ways to improve responsible work", "you often develop new ways to solve problems" and "you always present new ideas in the workplace". The scale of AV was adapted from Lindeman and Verkasalo [70] and included 9 items; for example, "you love living a challenging or exciting life", "you want to be accepted by society or respected by others" and "you want stability in your family life". Finally, the validation of brief measures of PT was adapted from Watson, et al. [71] and Sumalrot [72] and included 10 items; for example, "In general, how much do you like the work to be 'interesting', 'enthusiastic' and 'motivating'".

All scales were examined for content validity by 5 experts from the domains of organisational behaviour, management, industrial psychology, development of human resources and testing and evaluation. The experts had at least 10 years of experience in their respective fields, with a master's degree qualification or higher. Results revealed that content validity was between 0.8 and 1.00 and the confidence level of the questionnaire was 0.950.

Descriptive statistics were used to analyse the attributes of the samples and the variable levels, whereas reference statistics were used to examine the relationship among variables using SPSS 21.

Direct effects and moderators were analysed with PROCESS macro 3.1 in Model 3 [73]. This study analysed three-way interaction using Model 3 of Hayes [73].

### 3.2. Respond-Bias Control

To minimise inaccuracies resulting from the social response bias of the participants, several precautionary steps were taken following Podsakoff, et al. [74]. Firstly, the previously adopted questionnaire survey was back-translated into the mother tongue of the respondents. Brislin [75] considered that collecting data using the native language as a medium was more accurate. Secondly, to prevent social desirability bias as provision of the most pleasing answers, strict confidentiality was assured as suggested by Podsakoff, MacKenzie and Podsakoff [74]. Thirdly, Harman's single factor test was used to identify common method variance of the factors in line with Scott and Bruce [76]. Principal component analysis gave individual factors with 32.85% cumulative variance. Eichhorn [77] noted that Harman's single factor score for total variance at less than 50% suggested that common method bias (CMB) did not impact the data. Finally, the highest correlation (r) between variables AV and PT was between 0.271 to 0.665 and indicated that there was no respond bias, for which Spector and Brannick [78] stated that the correlation should be lower than 0.90 as a result of method respond bias.

### 4. Result

More than half of the samples were female (60.90%), while the rest were male (39.10%), with ages 26–30 (20.70%), 31–35 (18.70%) and more than 51 (6.60%). Regarding their educational levels, more than half graduated with a bachelor's degree (57.80%), followed by a master's degree (36.20%) and the remainder had doctoral degrees or lower than a bachelor's degree (6%). More than half of the samples had work experience of more than 10 years (53.20%), followed by 6–10 years (14.70%), with the remainder including less than 1 year, 2–3 years and 4–5 years (32.10%). Most of the samples were employees (49.10%), followed by managers/directors (25%) and senior officers and supervisors (25.90%).

The highest mean was found for AV, followed by PT, IWB and EE (4.331, 4.143, 4.115 and 3.793, respectively), as shown in Table 1. The highest standard deviation was EE, followed by IWB, PT and AV (0.896, 0.635, 0.538 and 0.426, respectively). Correlations among all variables were continuous. Correlations between independent variables and dependent variables were positive at 0.194 and 0.618, respectively. The multicollinearity was inspected before test hypothesis; multicollinearity was defined as that if two or more independent variables have an exact linear relationship between them then we have perfect multicollinearity. To detect the multicollinearity, there are high correlation coefficients, and pairwise correlations among independent variables might be high. If the correlation >0.8, then there is severe multicollinearity [79,80]. No pairs of variables were over 0.80, indicating no problems of multicollinearity.

**Table 1.** Means, standard deviations and correlation coefficients among study variables.

| Variable | Mean | S.D. | 1. | 2. | 3. | 4. |
|----------|------|------|------|------|------|------|
| 1. EE | 3.793 | 0.896 | 1.000 | | | |
| 2. AV | 4.331 | 0.426 | 0.194 ** | 1.000 | | |
| 3. PT | 4.143 | 0.538 | 0.293 ** | 0.605 ** | 1.000 | |
| 4. IWB | 4.115 | 0.635 | 0.322 ** | 0.511 ** | 0.618 ** | 1.000 |

Note: ** indicates correlation is significant at the 0.01 level (2-tailed). EE indicates employee engagement, AV indicates attainment value, PT indicates positive thinking; IWB indicates innovative work behaviour.

According to the size effect analysis in Table 2, EE positively affected IWB ($b = 0.155$, $p = 0.001$). This means that when individuals or employees had EE, they likely expressed their IWB. This finding confirmed hypothesis H1. Employees' perceived AV had moderating effects on the relationship between EE and IWB but with no statistical significance ($b = -0.113$, $p = 0.380$); this means hypothesis H2 was not confirmed. Meanwhile, PT showed moderating effects of AV and EE (Int_4) on IWB

with statistical significance ($t$ = 2.041, $b$ = 0.308, $p$ = 0.05 or 95% confidence interval without covering 0, or R2_change = 0.0055 resulting in F = 48.550, $p$ = 0.041 < 0.05); this means hypothesis H3 was confirmed. Thus, PT had co-effects with AV and EE on IWB as shown in Figure 2.

**Table 2.** Effect of innovative work behaviour (IWB) and moderator effect of attainment value (AV) and positive thinking (PT) on IWB.

| Variable | Coefficient | Standard Error | $t$-Value | $p$-Value | LLCI | ULCI |
|---|---|---|---|---|---|---|
| EE | 0.155 | 0.049 | 3.164 | 0.001 | 0.058 | 0.251 |
| PT | 0.755 | 0.091 | 8.303 | 0.000 | 0.576 | 0.934 |
| Int_1 | −0.156 | 0.094 | −1.658 | 0.098 | −0.341 | 0.029 |
| AV | 0.489 | 0.109 | 4.464 | 0.000 | 0.274 | 0.705 |
| Int_2 | −0.113 | 0.129 | −0.878 | 0.380 | −0.369 | 0.141 |
| Int_3 | 0.195 | 0.148 | 1.314 | 0.189 | −0.096 | 0.487 |
| Int_4 | 0.308 | 0.151 | 2.041 | 0.041 | 0.011 | 0.606 |
| $R^2$ = 0.669, $R^2$_change = 0.447, MSE = 0.561, F = 48.550, df1 = 7.000, df2 = 419, $p$ = 0.000, $R^2$_change = 0.0055 | | | | | | |

Note: Int_1 (Interaction 1): EE × PT, Int_2 (Interaction 2): EE × AV, Int_3 (Interaction 3): PT × AV, Int_4 (Interaction 4): EE × PT × AV, LLCI: lower levels for confidence interval, ULCI: upper levels for confidence interval.

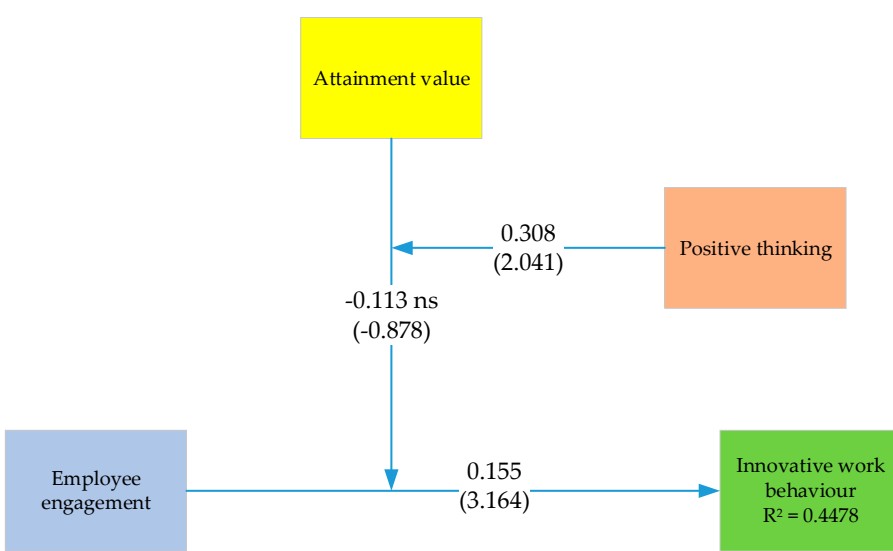

**Figure 2.** Moderating effects of AV and PT on employee engagement (EE) and IWB.

A pick-a-point analysis was conducted to determine nine points and detail the relationship between two moderators (PT and AV) on independent variables affecting dependent variables, as shown in Table 3.

1.  At lower PT (i.e., −0.538), when AV was at lower, middle and upper values (i.e., −0.426, 0.000, 0.426), how does EE affect IWB?
2.  At middle PT (i.e., 0.000), when AV was at lower, middle and upper values (−0.426, 0.000, 0.426), how does EE affect IWB?
3.  At upper PT (i.e., 0.5388), when AV was at lower, middle and upper values (−0.426, 0.000, 0.426), how does EE affect IWB?

**Table 3.** Simple slope values of moderators in the case of three-way interaction.

| MPT | MAV | Effect | Standard Error | *t*-Value | *p*-Value | LLCI | ULCI |
|---|---|---|---|---|---|---|---|
| −0.538 | −0.426 | 0.359 | 0.075 | 4.734 | 0.000 | 0.210 | 0.508 |
| −0.538 | 0.000 | 0.239 | 0.072 | 3.301 | 0.001 | 0.096 | 0.382 |
| −0.538 | 0.426 | 0.119 | 0.118 | 1.011 | 0.312 | −0.113 | 0.353 |
| 0.000 | −0.426 | 0.203 | 0.079 | 2.578 | 0.010 | 0.048 | 0.359 |
| 0.000 | 0.000 | 0.155 | 0.049 | 3.164 | 0.001 | 0.058 | 0.251 |
| 0.000 | 0.426 | 0.106 | 0.068 | 1.556 | 0.120 | −0.028 | 0.241 |
| 0.538 | −0.426 | 0.048 | 0.118 | 0.409 | 0.682 | −0.184 | 0.282 |
| 0.538 | 0.000 | 0.071 | 0.068 | 1.034 | 0.301 | −0.064 | 0.206 |
| 0.538 | 0.426 | 0.093 | 0.056 | 1.668 | 0.095 | −0.016 | 0.203 |

Note: MPT: mean of positive thinking, MAV: mean of attainment value, LLCI: lower levels for confidence interval, ULCI: upper levels for confidence interval.

The pick-a-point analysis resulted in statistical significance at some points. Four cases of the slope are shown in Table 3. In the first case, when PT is at the lower value, lower, middle and upper AV had moderating effects of EE on IWB ($b = 0.359$, 95% CI (0.210–0.508), $t = 4.734$, $p < 0.001$, $b = 0.239$, 95% CI (0.096–0.382), $t = 43.301$, $p < 0.001$ and $b = 0.119$, 95% CI (−0.113–0.353), $t = 1.011$, $p < 0.050$, respectively). In Figure 3, the equation line of low PT with low and middle AV had a different slope (i.e., effect or simple slope) from high AV. Thus, higher PT influenced AV to increase and resulted in a stronger relationship between EE and IWB indicated by the higher slope.

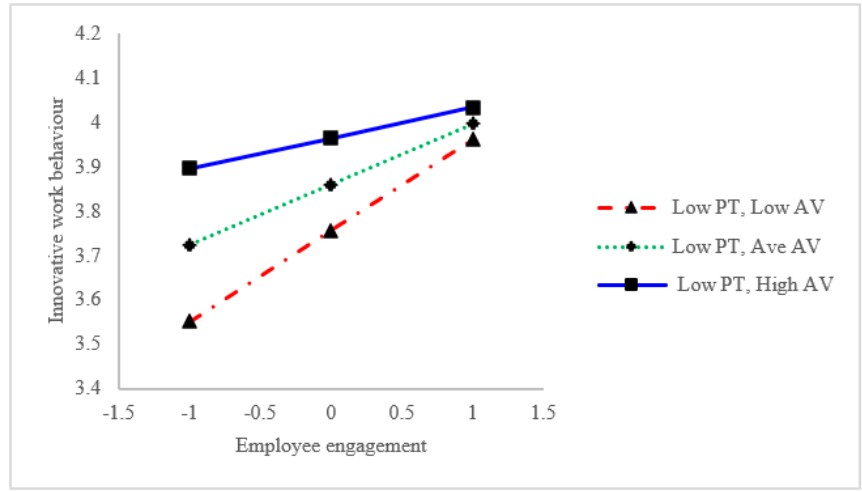

**Figure 3.** The lines present the relationship between EE and IWB for low PT with low, medium and high AV.

In the second case for middle PT, as shown in Figure 4, low, middle and high AV had moderating effects of EE on IWB ($b = 0.203$, 95% CI (0.048–0.359), $t = 2.578$, $p < 0.010$, $b = 0.155$, 95% CI (0.058–0.251), $t = 3.164$, $p < 0.001$ and $b = 0.106$, 95% CI (−0.028–0.241), $t = 1.556$, $p < 0.120$, respectively). In Figure 4, the equation lines of middle PT with low and middle AV had greater slopes (i.e., effect or simple slopes) than high AV. Thus, higher PT influenced AV to increase and resulted in a stronger relationship between EE and IWB.

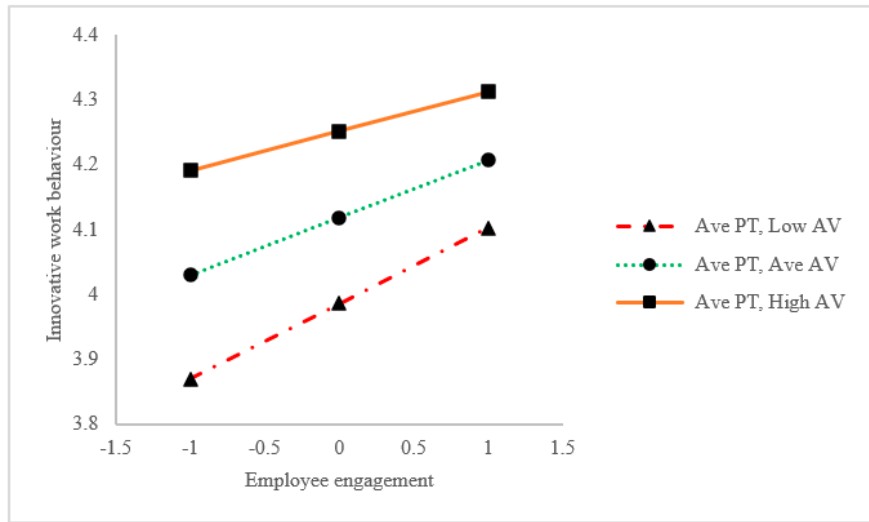

**Figure 4.** The lines present the relationship between EE and IWB for medium PT with low, medium and high AV.

In the third case with high PT, as shown in Figure 5, low, middle and high AV had moderating effects of EE on IWB ($b$ = 0.048, 95% CI (−0.184–0.282), $t$ = 0.409, $p$ < 0.682, $b$ = 0.071, 95% CI (−0.064–0.206), $t$ = 1.034, $p$ < 0.301 and $b$ = 0.093, 95% CI (−0.016–0.203), $t$ = 1.668, $p$ < 0.095, respectively). In Figure 4, the equation lines of low PT with low, middle and high AV had similar slopes (i.e., effect or simple slope). Thus, higher PT influenced AV to increase but with little effect as indicated by the similar slope.

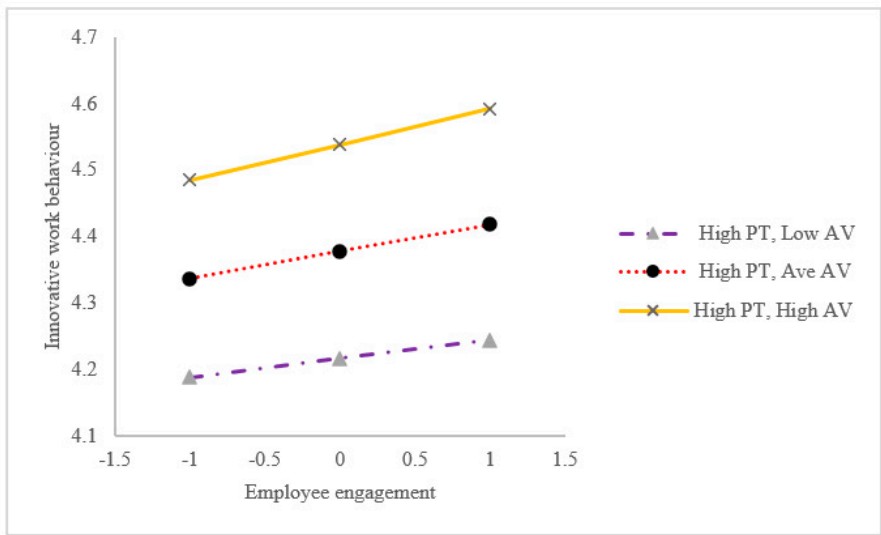

**Figure 5.** The lines present the relationship between EE and IWB for high PT with low, medium and high AV.

## 5. Discussion: Open Innovation from Employee Engagement with Attainment Value and Positive Thinking

Results supported Hypothesis 1, that EE influenced IWB. People with EE manifest their behaviours in three aspects: vigour, dedication and absorption. These aspects are expressed in the form of employees' behaviours and feelings for their work conditions rather than their assigned roles and responsibilities. They are also expressed as work ownership and desire to achieve company targets. Therefore, these people try to find guidelines or methods, problem solutions or new creative ways to meet company goals qualitatively and quantitatively [9,23–26,28,30]. Similarly, Slåtten and

Mehmetoglu [11] found that employees with work engagement were more determined to create new ways and methods to increase their work performance effectiveness.

Regarding Hypothesis 2, AV had no significant moderating effects between EE and IWB. This was not consistent with Hypothesis 1. However, PT variables had significant moderating effects of AV and EE on IWB (Hypothesis 3), consistent with Hypothesis 1. Hypothesis 3 concurred with the broaden-and-build theory. This explains forms and functions of PT that stimulate thoughts and feelings in a perceived attainment value. Three-way interactions demonstrated that the strongest positive relationship between EE and IWB occurred when low and medium PT were coupled with ether low and medium AV. The combined effects of low and middle PT and low and middle AV effects appear to enhance the EE and IWB relationship. Although research studies have shown that individuals may react more strongly to environmental stimuli when it is consistent with their inherent attainment value [16,60,62,63,81–85], this would help to explain the important of PT. However, it is possible that what drives the PT relationships is the appropriate level of the PT that gives impetus to higher reactivities.

In addition, employees with perceived AV and PT will promote employee engagement (EE) because both AV and PT are processes that are formed by positive perception and cognition as belief in possibilities based on reality, logic and generosity. Being determined, self-confident and able to control one's emotions affect employees' vigour, dedication and concentration to fulfil their responsibilities [62]. Moreover, a high level of employee engagement (EE) increases innovative work behaviour (IWB) because employees who are fully engaged in their work can unleash their full potential to attempt something new that will improve their performance. Moreover, employees who have high levels of innovative work behaviour (IWB) will try out new ideas and create alternative methods, services or products to benefit the company. Therefore, EE, AV and PT are set perimeter contribution factors for the development of innovation and discovery [86]. A company with open innovation will encourage and promote an outflow of knowledge to accelerate internal innovation and market expansion [87,88]. The concepts of EE, AV and PT are essential factors to activate IWB and positively influence the emergence of open innovation that can increase company prosperity [89,90].

## 6. Conclusions, Implication and Future Research Topics

### 6.1. Theoretical Implications

Results confirmed that EE had statistically significant effects on IWB, consistent with the concepts and theories of previous studies. Therefore, educators or researchers who are interested in studying IWB must also include EE concepts. However, previous studies only considered the direct effects of these two variables. This research fills the information gap by analysing and identifying the moderators co-affecting EE on IWB. Results showed that the moderators AV and PT co-predicted the variables at more than 44.78% with statistical significance. This implies the importance of these factors and educators and researchers should pay attention to AV and PT when explaining EE. EE is also necessary for explaining the phenomenon of IWB since EE is a psychological perception which manifests as dedication and work integration of employees [31,91–94]. When they possess work engagement, employees are ready to create innovative ways of working to make their jobs more effective. Meanwhile, if employees possess AV and PT, they will better engage with their work and this will further promote IWB [95]. If the perceived values of AV, PT and EE of employees are at low and middle levels, these aspects will strongly influence their behaviours.

### 6.2. Practical Implications

Managers can use these findings to explain the phenomenon of IWB stimulation from EE, AV and PT as co-moderators. Managers should encourage and support their workforce to build up EE by training, developing, reinforcing and supporting motivating activities. Moreover, managers should be supportive as motivators to promote EE by devoting due time and diligence to maximise the benefits

from their workforce [96]. When EE is applied in their work, employees will realise their potential responsibilities and be ready to express IWB.

Stimulating and supporting PT by employees is another important factor for managers to consider since PT correlates as a moderator between AV and EE. If managers stimulate PT by their employees, this will promote good attitudes and consciousness towards themselves and others. The employees will accept emerging problems with the strength to live happily and successfully [60,97,98]. Moreover, they will perceive the AV of their future careers, leading to greater EE with increased responsibilities.

### 6.3. Study Limitations and Future Research

This cross-sectional study covered a particular time period and explains the phenomenon occurring only within this time limit. Therefore, a longitudinal study should be conducted to collect data systematically in the long term to obtain a more complete dataset for further study with more reliable results. This model used EE as an independent variable, IWB as a dependent variable and AV and PT as moderators in the study context of HR officers in Thai society. Therefore, the model should be reinforced and extended to different contexts in terms of occupations, societies or cultures. The two moderators collaboratively change and predict IWB. Future research should be examined in relation to the relationship between EE and IWB in different contexts and examine other factors, such as motivation, leadership or perceived environment in the model, while experts in human resources or organisational behaviours should also be interviewed.

**Author Contributions:** Conceptualization, K.N.-N. and P.P.; Methodology, K.N.-N.; Software, N.W.; Validation, K.N.-N., P.P. and N.W.; Formal Analysis, K.N.-N..; Investigation, K.N.-N., P.P. and N.W.; Resources, P.P.; Data Curation, K.N.-N. and P.P.; Writing-Original Draft Preparation, K.N.-N.; Writing-Review & Editing, K.N.-N., P.P. and N.W.; Visualization, K.N.-N., P.P. and N.W.; Supervision, K.N.-N.; Project Administration, K.N.-N. and P.P. Funding Acquisition, P.P. All authors have read and agreed to the published version of the manuscript.

**Funding:** This research received no external funding.

**Conflicts of Interest:** The authors declare no conflict of interest.

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
