# Peer review of "Effect of Attainment Value and Positive Thinking as Moderators of Employee Engagement and Innovative Work Behaviour"

_2199-8531, doi:10.3390/joitmc6030069_

Round 1

Reviewer 1 Report

Introduction

The Introduction is definitely adequate, but could be improved.  This can be done by providing a little more extensive literature review of the constructs and stating more clearly what research need you are addressing and stating more clearly the contribution of your study.  Some more specific comments are as follows:

*In terms of the research need state more clearly in the Introduction that there is a need to examine moderators of the EE to IWB relationship. Doing will position better the importance of your paper early on.

*Define IWB and provide more research showing the importance of IWB (e.g. how it is related to organizational innovation and performance).

*Hypotheses 2-3:

Add “the relationship between” such as “Perceived attainment value moderates the relationship between EE and IWB” or” Perceived attainment value is a moderator between EE and IWB.”

*Add a Hypothesis 3 since your results focus on one direct effect and three different sets of interactions (AV solely, PT solely, and AV and PT together).

H3: Positive thinking moderates the relationship between EE and IWB

Then renumber H3 to H4

*Briefly state what the key postulates of the “broaden and build theory” are.

*P3: State what variables that AV moderates in relationship to employee work performance.  Clarify more clearly what you mean by “perception” in the reference to Purc and Laguna.  Provide a cite for the next sentence or rewrite it as a summary from the literature you cited previously in this paragraph.

*P3: State what variables PT acted as a moderator of (Khan and Husain).

  describing more what innovative work behaviors (IWB).   Note first well written and                                    

Research Methods

  1. 4 Provide more specifics on how you “improved” “applied” and “adapted” the various survey scales (e.g. what specific changes did you make to the survey questions and why did you make these changes)?
  2. 4 State more about the qualifications of the 5 experts who reviewed your scales.
  3. 4 Relabel “Respond” to “Response Bias”
  4. 5: Explain more the last sentence of the 1st paragraph – it is not clear to me in what regard you are mentioning r=.90.
  5. 5 Provide a cite for stating why no correlation over .80 indicates no problem of multicollinearity or delete this sentence.

State your results clearly in relation to your hypotheses in this section (e.g. these results support H1, etc.).

As noted above you should state support for a new H3 (that PT moderates the relationship of EE to IWB) and rewrite the end of the sentence referring to results with the statistics related to coeffects of AV and PT.

Be sure to specifically refer to Figures 2, 3, and 4 during the discussion of your results (e.g. in the paragraphs in which you discuss your results).

 All of 3 of your moderating effects include an ending sentence which states “However, (the) results showed that the case of low PT and high AV was still higher than the other cases”  - you should specifically list each time what you mean/define by the “other cases”                           

Discussion

Practical implications:  Provide more specifics on how managers/organizations can promote PT and AV in employees

Study limitations: You don’t need to state, “the results are not robust” and simply state that the study should be extended to different contexts, etc.  

Future Research:  State more specific moderating variables that should be examined in relation to the relationship between EE and IWB (e.g. the ones you suggested are pretty generic).

Writing

Overall, the writing is clear and the paper well organized.  There are some points that require minor editing:

Be sure to spell out the words the first time you use your abbreviations, such as IWB and PT (e.g. innovative work behaviors for IWB). 

Author Response

Dear Reviewers,

I am very appreciating that my manuscript has been reviewed and considered, thank you very much. All of the reviewer comments are very important to improve this paper appropriate for publication in “Journal of Open Innovation: Technology, Market, and Complexity — Open Access Journal”. Therefore, the 1st revised manuscript was carefully prepared and sent back for your consideration. Regarding to all questions, the corrections were defined in blue text with page number, please kindly find in the revision manuscript with tracked changes file. Any other additional point for the publication, please kindly again recommend. Lastly, I indeed hope that our correction would be satisfied and the paper will be published in the Journal of Open Innovation: Technology, Market, and Complexity — Open Access Journal. 

Sincerely Yours’

 The Corresponding author

Introduction

The Introduction is definitely adequate, but could be improved.  This can be done by providing a little more extensive literature review of the constructs and stating more clearly what research need you are addressing and stating more clearly the contribution of your study.  Some more specific comments are as follows:

*In terms of the research need state more clearly in the Introduction that there is a need to examine moderators of the EE to IWB relationship. Doing will position better the importance of your paper early on.

Action/Answer: Thank you so much for your kind suggestion. Revision has been made on page 2, line 49 to 61 as below:

Research results concurred with Kapikiran [18] and concluded that harmony emanates from positive thinking (PT) as an essential facet to regulate both personal and job satisfaction. Moreover, Chang, et al. [19] stated that focusing on PT promotes personal perspective as a starting point to respond to problems or issues. Adopting PT increases personal readiness for task implementation to meet the job requirements and attain the expected outcome.  The perceived difficulty of a task is conceptualised as a framework related to the cause and effect variables of the challenge and value of the job. Consideration of the task properties will determine the appropriate setting of goals and the enthusiasm to perform efficiency and successfully realise the objectives.  Arieli, et al. [20] and Purc and Laguna [13] highlighted key issues concerning attainment perception in relation to various personal innovative work behaviours, while Schwartz [21] considered that attainment perception could be used to accurately predict employee’s attainment value (AV).

To examine these concepts and theories, previous studies and recommendations from educators and researchers were collated with two main research objectives: i) to study the effects of EE on IWB and ii) to study AV and PA as moderators of EE and IWB.

*Define IWB and provide more research showing the importance of IWB (e.g. how it is related to organizational innovation and performance).

*Hypotheses 2-3:

Action/Answer: Thank you so much for your kind suggestion. Revision has been made on page 2 and 3, line 78 to 89 as below:

The concept of IWB refers to employees who aim to initiate new things and introduce new ideas useful for the production of products, services and new work procedures to achieve targets [32,33]. Similarly, Yeoh and Mahmood [34] stated that employees with IWB provide creative ideas concerning work procedures, methods, products and services. Improved work behaviour that attains targets leads to new initiatives and the introduction of novel ideas [35]. The expressive dimension of IWB closely relates to creative behaviour, especially by thinking up new useful ideas [36]. Nevertheless, IWB differs from creative thinking since it is aimed at applying new ideas that are useful for oneself, departments or organisations. 

Meanwhile, Caniëls and Veld [37] suggested that innovative work behaviour (IWB) had a significant statistical relationship with high work performance, whereas Bysted [38], Bos-Nehles, et al. [39],  Jol, et al. [40], Shanker, et al. [41] and Veenendaal and Bondarouk [42] revealed the significance of comparing the contexts of formal development. They found that individual intuition led to the creation and development of a strategic perspective that can be transferred to the work concept in groups and organisations.

Add “the relationship between” such as “Perceived attainment value moderates the relationship between EE and IWB” or” Perceived attainment value is a moderator between EE and IWB.”

*Add a Hypothesis 3 since your results focus on one direct effect and three different sets of interactions (AV solely, PT solely, and AV and PT together).

H3: Positive thinking moderates the relationship between EE and IWB

Then renumber H3 to H4

Action/Answer: Thank you so much for your kind suggestion. Revision has been made on page 4, line 143 to 152 as below:

       Khan and Husain [59] found that PT performs as a moderator in the relationship between independent variables and dependent variables with statistical significance, while Bandansin [57] indicated that PT was helpful for nurses to express their IWB and provide effective services for patients. These nurses integrated their existing knowledge with PT to form their IWB. As a result, the patients were increasingly impressed with their services or IWB. Supannopaph [58] mentioned that PT directs a person to perceive, behave and create positive results such as a healthy mind, opportunities for success, self-confidence, optimism and being realistic. Hazelton [60], Kang and Sung [61], Tufail, et al. [62] and  Young, et al. [63] conducted quantitative research and concluded that PT had a statistically significant relationship with EE. Their empirical findings supported that PT improved EE.

*Briefly state what the key postulates of the “broaden and build theory” are.

Action/Answer: Thank you so much for your kind suggestion. Revision has been made on page 4, line 125 to 131 as below:

This research examines and extends this assumption for a better explanation, understanding and restructuring following the broaden-and-build theory. This theory suggests that positive emotions (enjoyment, happiness, joy, interest and anticipation) broaden one’s awareness and encourage novel, varied and exploratory thoughts and actions. This broadened behavioural repertoire builds skills, while curiosity about a landscape becomes valuable navigational knowledge, pleasant interactions with a stranger become a supportive friendship and aimless physical play becomes exercise and physical excellence, expectation, interest, intuition, interpretation, integration and institutionalisation [15].

*P3: State what variables that AV moderates in relationship to employee work performance.  Clarify more clearly what you mean by “perception” in the reference to Purc and Laguna.  Provide a cite for the next sentence or rewrite it as a summary from the literature you cited previously in this paragraph.

*P3: State what variables PT acted as a moderator of (Khan and Husain).

Action/Answer: Thank you so much for your kind suggestion. Revision has been made on page 4, line 143 to 154 as below:

         Khan and Husain [59] found that PT performs as a moderator in the relationship between independent variables and dependent variables with statistical significance, while Bandansin [57] indicated that PT was helpful for nurses to express their IWB and provide effective services for patients. These nurses integrated their existing knowledge with PT to form their IWB. As a result, the patients were increasingly impressed with their services or IWB. Supannopaph [58] mentioned that PT directs a person to perceive, behave and create positive results such as a healthy mind, opportunities for success, self-confidence, optimism and being realistic. Hazelton [60], Kang and Sung [61], Tufail, et al. [62] and  Young, et al. [63] conducted quantitative research and concluded that PT had a statistically significant relationship with EE. Their empirical findings supported that PT improved EE.  According to these concepts, theories and previous research, the third hypothesis was posited as follows:

H3: positive thinking and attainment value combine as moderators in the relationship between employee engagement and innovative work behaviour.

  describing more what innovative work behaviors (IWB).   Note first well written

Action/Answer: Thank you so much for your kind suggestion. Revision has been made on page 2 and 3, line 78 to 89 as below:

The concept of IWB refers to employees who aim to initiate new things and introduce new ideas useful for the production of products, services and new work procedures to achieve targets [32-35]. Similarly, Yeoh and Mahmood [36] stated that employees with IWB provide creative ideas concerning work procedures, methods, products and services. Improved work behaviour that attains targets leads to new initiatives and the introduction of novel ideas [37]. The expressive dimension of IWB closely relates to creative behaviour, especially by thinking up new useful ideas [38]. Nevertheless, IWB differs from creative thinking since it is aimed at applying new ideas that are useful for oneself, departments or organisations. 

Meanwhile, Caniëls and Veld [39] suggested that innovative work behaviour (IWB) had a significant statistical relationship with high work performance, whereas Bysted [40], Bos-Nehles, et al. [41],  Jol, et al. [42], Shanker, et al. [43] and Veenendaal and Bondarouk [44] revealed the significance of comparing the contexts of formal development. They found that individual intuition led to the creation and development of a strategic perspective that can be transferred to the work concept in groups and organisations.

Research Methods

 Provide more specifics on how you “improved” “applied” and “adapted” the various survey scales (e.g. what specific changes did you make to the survey questions and why did you make these changes)?

Action/Answer: Thank you so much for your kind suggestion. Revision has been made on page  5 line 177 to 188 as below:

This research design followed a comparative longitudinal case study approach as the most appropriate method to identify the main sequences of sampling population and culture for different units of analysis in the Thai context. Data collection for EE was adapted from  Saks [27] and comprised 6 items including “you feel that the job is part of your life”, “although the work takes a lot of time to complete you like to do it” and “you tend to focus on the work without thinking of anything else”. Data collection for IWB was adapted from Janssen [32] and included 4 items, for example “you often develop new ways to improve responsible work”, “you often develop new ways to solve problems” and “you always present new ideas in the workplace”. The scale of AV was adapted from Lindeman and Verkasalo [66] and included 9 items, for example “you love living a challenging or exciting life”, “you want to be accepted by society or respected by others” and “you want stability in your family life”. Finally, the validation of brief measures of PT was adapted from Watson, et al. [67] and Sumalrot [68] and included 10 items, for example “In general, how much do you like the work to be ‘interesting’, ‘enthusiastic’ and ‘motivating’”.

4 State more about the qualifications of the 5 experts who reviewed your scales.

Action/Answer: Thank you so much for your kind suggestion. Revision has been made on page  5, line 189to 191 as below:

All scales were examined for content validity by 5 experts from the domains of organisational behaviour, management, industrial psychology, development of human resources and testing and evaluation. The experts had at least 10 years of experience in their respective fields with a master’s degree qualification or higher.

Relabel “Respond” to “Response Bias”

Action/Answer: Thank you so much for your kind suggestion. Revision has been made on page  5 and 6, line 199 to 209 as below:

To minimise inaccuracies resulting from the social response bias of the participants, several precautionary steps were taken following Podsakoff, et al. [70]. Firstly, the previously adopted questionnaire survey was back-translated into the Mother tongue of the respondents. Brislin [71] considered that collecting data using the native language as a medium was more accurate. Secondly, to prevent social desirability bias as provision of the most pleasing answers, strict confidentiality was assured as suggested by Podsakoff, MacKenzie and Podsakoff [70]. Thirdly, Harman’s single factor test was used to identify common method variance of the factors in line with Scott and Bruce [72]. Principal component analysis gave individual factors with 32.848% cumulative variance. Eichhorn [73] noted that Harman’s single factor score for total variance at less than 50% suggested that common method bias (CMB) did not impact the data.

5: Explain more the last sentence of the 1st paragraph – it is not clear to me in what regard you are mentioning r=.90.

Action/Answer: Thank you so much for your kind suggestion. Revision has been made on page  6, line 207 to 209 as below:

Finally, the highest correlation (r) between variables AV and PT was between 0.271 to 0.665 indicated that there was no respond bias which Spector and Brannick [74] and [74] stated that the correlation should be lower than 0.90 as a result of method respond bias.

Provide a cite for stating why no correlation over .80 indicates no problem of multicollinearity or delete this sentence.

Action/Answer: Thank you so much for your kind suggestion. Revision has been made on page  6, line 226 to 232 as below:

Correlations among all variables were continuous. Correlations between independent variables and dependent variables were positive at 0.194 and 0.618 respectively. The multicollinearity was inspected before test hypothesis, multicollinearity defined that if two or more independent variables have an exact linear relationship between them then we have perfect multicollinearity. To detection the multicollinearity there is high correlation coefficients pairwise correlations among independent variables might be high. If the correlation > 0.8 then severe multicollinearity. No pairs of variables were over 0.80, indicating no problems of multicollinearity.  

State your results clearly in relation to your hypotheses in this section (e.g. these results support H1, etc.).

Action/Answer: Thank you so much for your kind suggestion. Revision has been made on page  6, line 233 to 235 as below:

According to the size effect analysis in Table 2, EE positively affected IWB (b = 0.155, p = 0.001). This means that when individuals or employees had EE, they likely expressed their IWB, This finding confirmed hypothesis H1.

As noted above you should state support for a new H3 (that PT moderates the relationship of EE to IWB) and rewrite the end of the sentence referring to results with the statistics related to coeffects of AV and PT.

Action/Answer: Thank you so much for your kind suggestion. Revision has been made on page  6, line 233 to 239 as below:

According to the size effect analysis in Table 2, EE positively affected IWB (b = 0.155, p = 0.001). This means that when individuals or employees had EE, they likely expressed their IWB, This finding confirmed hypothesis H1. Employees’ perceived AV had moderating effects on the relationship between EE and IWB but with no statistical significance (b = -0.113, p = 0.380) this means hypothesis H2 was not confirmed, while PT showed moderating effects of AV and EE (Int_4) on IWB with statistical significance (t = 2.041, b = 0.308, p = 0.05 or 95% confidence interval without covering 0, or R2_change = 0.0055 resulting in F = 48.550, p = 0.041 < 0.05) this means hypothesis H3 was confirmed.

Be sure to specifically refer to Figures 2, 3, and 4 during the discussion of your results (e.g. in the paragraphs in which you discuss your results).

Action/Answer: Thank you so much for your kind suggestion. Revision has been made on page  8, line 273 to 274, 280 and 288 as below:

In Figure 2, the equation line of low PT with low and middle AV had a different slope (i.e. effect or simple slope) from high AV.

In the second case for middle PT (Figures 3),

In the third case with high PT (Figures 4),

All of 3 of your moderating effects include an ending sentence which states “However, (the) results showed that the case of low PT and high AV was still higher than the other cases”  - you should specifically list each time what you mean/define by the “other cases”       

Action/Answer: Thank you so much for your kind suggestion. To avoid ambiguous, we decided all the sentence However, results showed that the case of low PT and high AV was still higher than the other cases at the end paragraph.           

Discussion

Practical implications:  Provide more specifics on how managers/organizations can promote PT and AV in employees

Action/Answer: Thank you so much for your kind suggestion. Revision has been made on page 10, line 331 to 340 as below:

Managers can use these findings to explain the phenomenon of IWB stimulation from EE, AV and PT as co-moderators. Managers should encourage and support their workforce to build up EE by training, developing, reinforcing and supporting motivating activities. Moreover, managers should be supportive as motivators to promote EE by devoting due time and diligence to maximise the benefits from their workforce. When EE is applied in their work, employees will realise their potential responsibilities and be ready to express IWB. 

Stimulating and supporting PT by employees is another important factor for managers to consider since PT correlates as a moderator between AV and EE. If managers stimulate PT by their employees this will promote good attitudes and consciousness towards themselves and others. The employees will accept emerging problems with the strength to live happily and successfully [45]. Moreover, they will perceive the AV of their future careers, leading to greater EE with increased responsibilities. 

Study limitations: You don’t need to state, “the results are not robust” and simply state that the study should be extended to different contexts, etc.  

Thank you for your kind suggestion. Revision has been revised

Action/Answer: Thank you so much for your kind suggestion. Revision has been made on page 11, line 347 to 351 as below:

Therefore, the model should be reinforced and extended to different contexts in terms of occupations, societies or cultures. The two moderators collaboratively change and predict IWB. Future research should examine other factors such as motivation, leadership or perceived environment, while experts in human resources or organisational behaviours should also be interviewed.  

Future Research:  State more specific moderating variables that should be examined in relation to the relationship between EE and IWB (e.g. the ones you suggested are pretty generic).

Action/Answer: Thank you so much for your kind suggestion. Revision has been made on page 11, line 348 to 351 as below:

Future research should be examined in relation to the relationship between EE and IWB in difference contexts and examine other factors such as motivation, leadership or perceived environment in the model, while experts in human resources or organisational behaviours should also be interviewed.  

Reviewer 2 Report

The article deals with roles of attainment value and positive thinking as moderators of employee engagement and innovative work behavior.

The study is relatively properly organized. But, lacking of proper references and proper fits to the theme of the journal (JOI) must be improved.

I recommend for the author(s) to make the article to be improved following below suggestions.

  1. Please search and find proper references from JOI and special issues of TFSC, STS, EPS, Sustainability, those deals with open innovation (OI) as their special issue theme. 
  2. By doing 1., please find relations of the article to the theme of open innovation (OI), and please make some meaningful discussions about that.
  3. In the implications part, if possible, please mention actual case(s) that can explain the results ad implications of the study.
  4. Sample descriptions are not much concrete. Please make it more concrete and detailed for the sample.   

Author Response

Dear Reviewers,

I am very appreciating that my manuscript has been reviewed and considered, thank you very much. All of the reviewer comments are very important to improve this paper appropriate for publication in “Journal of Open Innovation: Technology, Market, and Complexity — Open Access Journal”. Therefore, the 1st revised manuscript was carefully prepared and sent back for your consideration. Regarding to all questions, the corrections were defined in blue text with page number, please kindly find in the revision manuscript with tracked changes file. Any other additional point for the publication, please kindly again recommend. Lastly, I indeed hope that our correction would be satisfied and the paper will be published in the Journal of Open Innovation: Technology, Market, and Complexity — Open Access Journal. 

The article was proofread and edited by native Mr. Peter from England.

Sincerely Yours’

    The Corresponding author

Writing

Overall, the writing is clear and the paper well organized.  There are some points that require minor editing:

Be sure to spell out the words the first time you use your abbreviations, such as IWB and PT (e.g. innovative work behaviors for IWB). 

Action/Answer: Thank you so much for your kind suggestion. Revision has been made as below:

Innovative work behaviour (IWB) shows at the first sentence on page 1 line 25

Employee engagement (EE) shows at first time on page 1 line 34

attainment value (AV) shows at first time on page 2 line 41

positive thinking (PT) shows at first time on page 2 line 43

The article deals with roles of attainment value and positive thinking as moderators of employee engagement and innovative work behavior. The study is relatively properly organized. But, lacking of proper references and proper fits to the theme of the journal (JOI) must be improved.

I recommend for the author(s) to make the article to be improved following below suggestions.

 Please search and find proper references from JOI and special issues of TFSC, STS, EPS, Sustainability, those deals with open innovation (OI) as their special issue theme. 

By doing 1., please find relations of the article to the theme of open innovation (OI), and please make some meaningful discussions about that.

Action/Answer: Thank you so much for your kind suggestion. The articles from JOI has been cited in the paper as below:

Liu, Y.; Kim, J.; Yoo, J. Intangible Resources and Internationalization for the Innovation Performance of Chinese High-Tech Firms. Journal of Open Innovation: Technology, Market, and Complexity 2019, 5, 52.

Lee, J.; Kim, D.; Sung, S. The Effect of Entrepreneurship on Start-Up Open Innovation: Innovative Behavior of University Students. Journal of Open Innovation: Technology, Market, and Complexity 2019, 5, 103.

Kimseng, T.; Javed, A.; Jeenanunta, C.; Kohda, Y. Applications of Fuzzy Logic to Reconfigure Human Resource Management Practices for Promoting Product Innovation in Formal and Non-Formal R&D Firms. Journal of Open Innovation: Technology, Market, and Complexity 2020, 6, 38.

Setini, M.; Yasa, N.N.K.; Gede Supartha, I.W.; Ketut Giantari, I.; Rajiani, I. The Passway of Women Entrepreneurship: Starting from Social Capital with Open Innovation, through to Knowledge Sharing and Innovative Performance. Journal of Open Innovation: Technology, Market, and Complexity 2020, 6, 25.

Vrchota, J.; MaÅ™iková, M.; ŘehoÅ™, P.; Rolínek, L.; Toušek, R. Human Resources Readiness for Industry 4.0. Journal of Open Innovation: Technology, Market, and Complexity 2020, 6, 3.

In the implications part, if possible, please mention actual case(s) that can explain the results ad implications of the study.

Action/Answer: Thank you so much for your kind suggestion. Revision has been made on page 10, line 331 to 340 as below:

Managers can use these findings to explain the phenomenon of IWB stimulation from EE, AV and PT as co-moderators. Managers should encourage and support their workforce to build up EE by training, developing, reinforcing and supporting motivating activities. Moreover, managers should be supportive as motivators to promote EE by devoting due time and diligence to maximise the benefits from their workforce. When EE is applied in their work, employees will realise their potential responsibilities and be ready to express IWB. 

Stimulating and supporting PT by employees is another important factor for managers to consider since PT correlates as a moderator between AV and EE. If managers stimulate PT by their employees this will promote good attitudes and consciousness towards themselves and others. The employees will accept emerging problems with the strength to live happily and successfully [45]. Moreover, they will perceive the AV of their future careers, leading to greater EE with increased responsibilities. 

Sample descriptions are not much concrete. Please make it more concrete and detailed for the sample.   

Action/Answer: Thank you so much for your kind suggestion. Revision has been made on page 4 and 5, line 157 to 171 as below:

Analyzing units of this research included human resource (HR) officers who were members of the Personnel Management Association of Thailand. Selection of HR officers as the targeted population was appropriate because they were expected to express IWB to solve various problems and manage and develop effective employees. HR officers usually possess EE because they work with people who have various problems; without EE they cannot perform their work effectively. HR officers also know their career paths and targets. Therefore, they are people with perceived AV which companies push their employees to possess. PT is an important factor when working with other people as well as for performing responsibilities dealing with complicated and challenging problems. Therefore, HR officers must have positive attitudes towards their responsibilities and interactions with other employees.

 The sample size included 400 sampling units. A convenience sampling method was chosen using email correspondence as the most effective way to support the variety and quality of the respondents. Online surveys have both advantages and disadvantages but responses can be obtained quickly without geographical limitation. The email surveys were sent to 400 selected samples. Within 4 weeks, 253 surveys were returned so another 147 email surveys were sent out to a new group of samples and 95 responding emails were obtained within a similar waiting period of 4 weeks.
